# Evidence of exposure to SARS-CoV-2 in cats and dogs from households in Italy

E. I. Patterson [1], G. Elia[2], A. Grassi[3], A. Giordano[4], C. Desario[2], M. Medardo[5], S. L. Smith[6], E. R. Anderson[1], T. Prince[7], G. T. Patterson [6], E. Lorusso[2], M. S. Lucente[2], G. Lanave[2], S. Lauzi[4], U. Bonfanti[5], A. Stranieri[4], V. Martella[2], F. Solari Basano[8], V. R. Barrs[9], A. D. Radford[6], U. Agrimi[10], G. L. Hughes [1], S. Paltrinieri [4] & N. Decaro [2 ✉]

SARS-CoV-2 emerged from animals and is now easily transmitted between people. Sporadic detection of natural cases in animals alongside successful experimental infections of pets, such as cats, ferrets and dogs, raises questions about the susceptibility of animals under natural conditions of pet ownership. Here, we report a large-scale study to assess SARS-CoV-2 infection in 919 companion animals living in northern Italy, sampled at a time of frequent human infection. No animals tested PCR positive. However, 3.3% of dogs and 5.8% of cats had measurable SARS-CoV-2 neutralizing antibody titers, with dogs from COVID-19 positive households being significantly more likely to test positive than those from COVID-19 negative households. Understanding risk factors associated with this and their potential to infect other species requires urgent investigation.

[1] Centre for Neglected Tropical Disease, Departments of Vector Biology and Tropical Disease Biology, Liverpool School of Tropical Medicine, Pembroke Place, Liverpool L3 5QA, UK. [2] Department of Veterinary Medicine, University of Bari Aldo Moro, Strada Prov. per Casamassima Km 3, 70010 Valenzano, BA, Italy. [3] I-VET srl, Laboratorio di Analisi Veterinarie, Via Ettore Majorana, 10 - 25020 Flero, BS, Italy. [4] Department of Veterinary Medicine, Veterinary Teaching Hospital, University of Milan, Via dell'Università 6, 26900 Lodi, Italy. [5] La Vallonèa Veterinary Diagnostic Laboratory, via G. Sirtori 9, 20017 Passirana di Rho, MI, Italy. [6] Institute of Infection, Veterinary and Ecological Sciences, University of Liverpool, Leahurst Campus, Chester High Road, Neston CH64 7TE, UK. [7] NIHR Health Protection Unit in Emerging and Zoonotic Infections, Department of Clinical Infection, Microbiology and Immunology, University of Liverpool, Liverpool, UK. [8] Arcoblu s.r.l., via Alessandro Milesi 5, 20133 Milan, Italy. [9] City University's Jockey Club College of Veterinary Medicine and Life Sciences, 5/F, Block 1A, To Yuen Building, 31 To Yuen Street, Kowloon, Hong Kong. [10] Department of Food Safety, Nutrition and Veterinary Public Health, Istituto Superiore di Sanità, Viale Regina Elena, 299, 00161 Rome, Italy. ✉email: nicola.decaro@uniba.it

Cases of severe acute respiratory syndrome coronavirus 2 (SARS-CoV-2) infection were detected in late December 2019 in Wuhan, Hubei Province, China[1], possibly as a spillover from bats to humans[2], and rapidly spread worldwide becoming a pandemic[3]. Although the virus is believed to spread almost exclusively by human-to-human transmission, there are concerns that some animal species may contribute to the ongoing SARS-CoV-2 pandemic[4]. To date, sporadic cases of SARS-CoV-2 infection have been reported in dogs and cats. These include detection of SARS-CoV-2 RNA in respiratory and/or fecal specimens of dogs and cats with or without clinical signs[5–7], as well as of specific antibodies in sera from pets from coronavirus disease 2019 (COVID-19)-affected areas[7,8]. In addition, experimental infection of various animal species has demonstrated that although dogs appear poorly susceptible to SARS-CoV-2 infection, developing asymptomatic infections and shedding low titer or no virus, cats develop respiratory pathology and shed high titers of SARS-CoV-2, even being able to infect in-contact animals[9,10]. Wide-scale testing of susceptible species is needed to assess the extent of animal infection under more natural conditions of husbandry. Here we conducted an extensive epidemiological survey from March to May 2020 in cats and dogs living in Italy, either in SARS-CoV-2-positive households or living in geographic areas that were severely affected by COVID-19. To our knowledge, this is the largest study to investigate SARS-CoV-2 in companion animals to date.

## Results

All animals were sampled by their private veterinary surgeon during routine healthcare visits between 15 March and 11 May 2020 (Source data S1). Sampling of animals for this study was approved by the Ethics Committee of the Department of Veterinary Medicine, University of Bari, Italy (approval number 15/2020). A total of 603 dogs and 316 cats were sampled from different Italian regions, mostly Lombardy (476 dogs, 187 cats). Animals were sampled either from regions severely affected by COVID-19 outbreaks in humans or from those that offered convenient access to samples. Oropharyngeal (303 dogs, 173 cats), nasal (183 dogs, 78 cats), and/or rectal (66 dogs, 30 cats) swabs were collected from a total of 494 pets using synthetic fiber swabs (Table 1). For 340 dogs and 188 cats, full signalment and clinical history were collected, including breed, sex, age, exposure to COVID-19-infected humans in the previous 2 weeks (COVID-19-positive household, suspected COVID-19-positive household but not confirmed by specific assay, and COVID-19-negative household), and the presence of respiratory signs (cough, sneezing, conjunctivitis, nasal and/or ocular discharge). Pets living with SARS-CoV-2-infected patients included 47 dogs and 22 cats for serology and 64 dogs and 57 cats for molecular investigations,

with a single animal being sampled from each COVID-19-positive household.

Sera were available for 188 dogs and 63 cats for which complete signalment, history, and location were available (Fig. 1). Additional sera were collected from diagnostic laboratories for 263 dogs and 128 cats from the affected areas, but which lacked further historical information.

Detection of SARS-CoV-2 RNA used two real-time reverse-transcription PCR (RT-PCR) assays targeting nucleoprotein and envelope protein genes[11]. Plaque reduction neutralization tests (PRNTs)[12] were performed with SARS-CoV-2/human/Liverpool/REMRQ0001/2020 isolate[13]. $PRNT_{80}$ was determined by the highest dilution with ≥80% reduction in plaques compared to the control.

All 494 animals from which at least a swab was available tested negative for SARS-CoV-2 RNA, including 38 cats and 38 dogs that showed respiratory disease at the time of sampling, suggesting the absence of active SARS-CoV-2 infection in the tested animals. In addition, 64 of these dogs and 57 of the cats that tested negative were living in households previously confirmed as having had COVID-19 and 14 animals (11 cats and 3 dogs) from COVID-19 households were displaying respiratory signs at the time of sampling.

SARS-CoV-2-neutralizing antibodies were detected in 15 dogs (3.3%, 15/451) and 11 cats (5.8%, 11/191), with titers ranging from 1:20 to 1:160 and from 1:20 to 1:1280 in dogs and cats, respectively. Of samples from households with known COVID-19 status, neutralizing antibodies were detected in 6 of 47 dogs (12.8%) and 1 of 22 cats (4.5%) from COVID-19-positive households, 1 of 7 dogs (14.3%) and 0 of 3 cats (0%) from suspected COVID-19-positive households, and 2 of 133 dogs (1.5%) and 1 of 38 cats (2.6%) from COVID-19-negative households (Table 2). For those 423 animals where an age was recorded, 0 of 30 aged <1 year (0.0%), 6 of 92 aged 1–3 years (6.5%), 3 of 102 aged 4–7 years (2.9%), and 6 of 199 aged 8 years and over (3.0%) tested positive. None of the animals with neutralizing antibodies displayed respiratory signs at the time of sampling.

Reference sera or ascitic fluids from animals previously shown to be positive for canine enteric coronavirus[14] (three sera), canine respiratory coronavirus[15] (three sera), and feline coronavirus[16] (three ascitic fluids) tested negative by the PRNT assay for SARS-CoV-2, confirming the specificity of the obtained results[8].

Dogs were significantly more likely to test positive for SARS-CoV-2-neutralizing antibodies if they came from a known COVID-19-positive household (Fisher's exact test, $p = 0.004$) or were male (Fisher's exact test, $p = 0.045$), whereas there was insufficient data to assess any correlation with the neuter status, as this was reported only for 1 seropositive bitch. For provinces in the Lombardy region where at least ten samples were available, there was a positive trend between the proportion of dogs that tested positive and the recorded burden of human disease (Spearman's $r = 0.771$, $p = 0.103$) (Fig. 2) (Source data S2). A similar association was observed for cats (Spearman's $r = 0.696$, $p = 0.125$).

## Discussion

Following its original probable transmission to humans from animals, SARS-CoV-2 has spread globally within the human population with devastating health and economic impacts. To date, SARS-CoV-2 has been sporadically detected in naturally infected dogs and cats, most of which were living in close contact with infected humans. Most studies of companion animals are small in nature, likely because of an inevitable research focus on human disease. Our results from this extensive study of SARS-CoV-2 infection in owned pets living in areas where viral

**Table 1 Sample type distribution for detection of SARS-CoV-2 RNA.**

| Sample type | Dogs | Cats | Total |
|---|---|---|---|
| OP only | 131 | 98 | 229 |
| N only | 11 | 3 | 14 |
| R only | 0 | 0 | 0 |
| OP + N | 106 | 49 | 155 |
| OP + R | 0 | 4 | 4 |
| N + R | 0 | 0 | 0 |
| O + N + R | 66 | 26 | 92 |
| Total | 314 | 180 | 494 |

*N* nasal swab, *OP* oropharyngeal swab, *R* rectal swab.

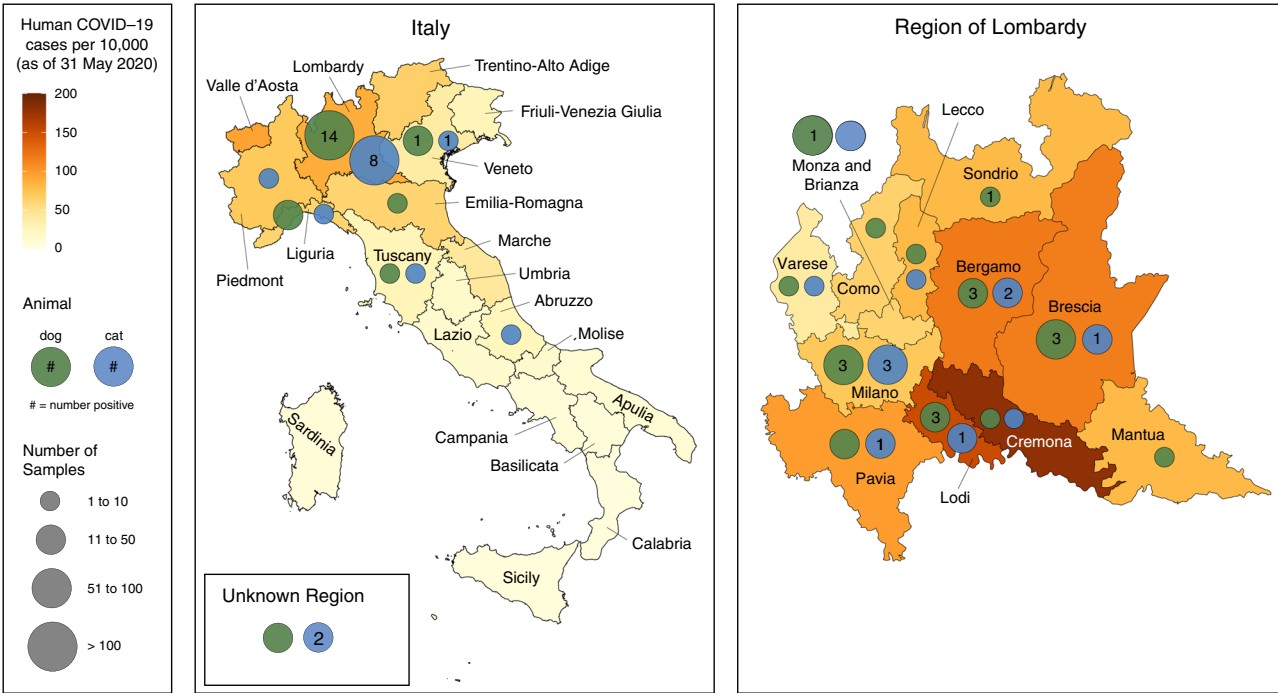

**Fig. 1 Distribution of dog and cat samples assayed for neutralizing antibody titer across Italy and the region of Lombardy.** Data on human COVID-19 cases from the Italian Department of Civil Protection as of 31 May 2020 and population data from the Italian National Institute of Statistics (ISTAT), January 2019. Source data are provided as a Source Data file.

**Table 2 Seropositivity among dogs and cats, split into risk factor groupings where data were available[a].**

| Risk factor | Dogs | | | Cats | | |
|---|---|---|---|---|---|---|
| | No. +(total) | % | p | No. +(total) | % | p |
| Household | | | 0.004 | | | 1.000 |
| Covid+ | 6 (47) | 12.8% | | 1 (22) | 4.5% | |
| Covid− | 2 (133) | 1.5% | | 1 (38) | 2.6% | |
| Suspected Covid+ | 1 (7) | 14.3% | | 0 (3) | 0.0% | |
| Sex | | | 0.045 | | | 0.492 |
| Male | 7 (83) | 8.4% | | 2 (31) | 6.5% | |
| Female | 2 (105) | 1.9% | | 0 (30) | 0.0% | |
| Age (years) | | | na | | | na |
| <1 | 0 (20) | 0.0% | | 0 (9) | 0.0% | |
| 1–3 | 5 (70) | 7.1% | | 1 (22) | 4.5% | |
| 4–7 | 2 (83) | 2.4% | | 1 (19) | 5.3% | |
| 8+ | 4 (137) | 2.9% | | 2 (62) | 3.2% | |
| Unknown | 4 (141) | 2.8% | | 7 (78) | 9.0% | |

[a]For household and sex, p-value determined by two-sided Fisher's exact test. Household COVID + defined as one or more members of a household with a confirmed positive COVID-19 test. All the information was not available for all the animals. Both household (p = 0.004) and sex (p = 0.045) were associated with COVID seropositivity among dogs, whereas neither household (p = 1.000) nor sex (p = 0.492) were associated with COVID seropositivity among cats.

transmission was active in the human population confirms field observations that both cats and dogs can seroconvert under the normal conditions of pet ownership.

The link between SARS-CoV-2 household infection and a pet's seropositivity was only apparent for dogs, possibly suggesting greater interaction between positive people and their household dogs as compared to cats. This contrasts experimental studies where dogs were less susceptible to infection[9]. However, this finding could be also reflective of the small numbers of

seropositive cats identified. In addition, a higher proportion of male dogs were seropositive compared to female dogs, but also in this case the results could have been biased by the small size of positive animals. In fact, neuter status was unknown for the majority of seropositive animals, which prevented a full comparison with the situation in humans. Future studies in animals and humans should investigate whether this phenomenon is based in physiological or behavioral differences between males and females. Although there are clear gender differences in outcomes in human COVID-19 infections, with males at higher risk of severe disease, there seems to be no evidence for a difference in infection risk[17]. None of the 30 juvenile animals, <1 year of age, tested positive. Our findings are consistent with reports of other seropositive naturally exposed cats and dogs, which were all adult[6,7], and in experimentally infected adult cats and dogs where clinical signs were not displayed[18]. However, in cats aged <1 year a higher susceptibility to SARS-CoV-2 infection was observed at 70–100 days than at 8 months of age[9].

In contrast to the serology results, all animals tested negative by PCR, including those animals living in households with confirmed COVID-19 human infection and those with and without respiratory signs. This suggests that although pet animals can seroconvert, they may shed virus for relatively short periods of time. In experimental studies, cats stopped shedding virus by 10 days post infection (dpi) and developed neutralizing antibody responses by 7–13 dpi[9,18]. Similar results were reported in experimental infection of dogs, in which virus was detected in feces up to 6 dpi, but not in oropharyngeal swabs[6]. However, in a naturally infected Pomeranian dog, SARS-CoV-2 RNA was detected from nasal swabs by quantitative RT-PCR for at least 13 days at low titer, whereas the virus was not detected in fecal/rectal samples[7], suggesting that virus shedding patterns may vary in some animals. Half of the challenged dogs had detectable antibodies by 14 dpi. These studies and our own highlight similar challenges in detecting SARS-CoV-2 infection that exist for both humans and animals[19]. It is not possible with our field data set to

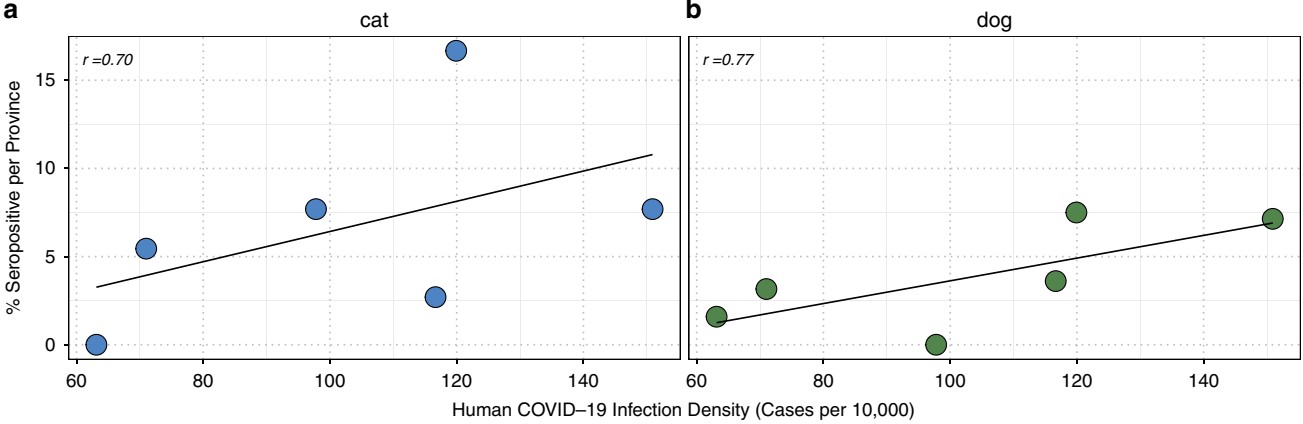

**Fig. 2 Correlation of percentage of seropositive animals per province and human COVID-19 infection density in Lombardy.** Data points were taken from provinces with at least ten samples. Spearman's correlation was used to assess association. Source data are provided as a Source Data file.

estimate the time of infection in animals that were seropositive, and restrictions on human and animal movement during the pandemic may have delayed visits to veterinary practitioners where sampling occurred. We advocate the inclusion of pets in ongoing assessments of community and household shedding to improve detection of active infection.

In this extensive epidemiological survey of SARS-CoV-2, we found that companion animals living in areas of high human infection have been exposed to SARS-CoV-2, thus confirming previous reports of natural infections of dogs and cats with the novel coronavirus. Our results suggest that dogs warrant further investigation regarding SARS-CoV-2 susceptibility, although higher antibody titers were detected in cats, which is in agreement with previous studies, suggesting that these animals are most susceptible to SARS-CoV-2 infection[9,18]. We also observed seropositivity rates in animals comparable to those of humans via community sampling at a similar time in European countries[20–22]. This suggests that infection in companion animals is not unusual. Based on current knowledge, it is unlikely that infected pets play an active role in SARS-CoV-2 transmission to humans. However, animal-to-human transmission may be more likely under certain environmental conditions, such as the high animal population densities encountered on infected mink farms[23]. As and when human transmission becomes rarer and contact tracing becomes more accessible, serological surveillance of pets may be advocated to develop a holistic picture of community disease dynamics and ensure that all transmission opportunities are terminated.

## Methods

**Samples.** All animals were sampled by their private veterinary surgeon during a healthcare visit for other reasons. A total of 603 dogs and 316 cats were sampled from different Italian regions, mostly Lombardy (476 dogs, 187 cats). Animals were sampled either from regions severely affected by COVID-19 outbreaks in humans or from those that offered convenient access to samples. Oropharyngeal (303 dogs, 173 cats), nasal (183 dogs, 78 cats), and/or rectal (66 dogs, 30 cats) swabs were collected from a total of 494 pets using synthetic fiber swabs (Table 1). Oropharyngeal swabs were collected by inserting the swab into the posterior pharynx and tonsillar areas. For collection of nasal secretions, the same swab was inserted into each nostril and rotated in order to be saturated by the nasal fluid. Rectal swabs were collected by inserting the swab 1–2 cm past the anal verge and rotating the swab gently 360°. After swabbing each swab was immersed in 1.5 ml of viral transport medium.

For 340 dogs and 188 cats, full signalment and clinical history were available, including breed, sex, age, exposure to COVID-19-infected humans (COVID-19-positive household, suspected COVID-19-positive household but not confirmed by specific assay, and COVID-19-negative household), the presence of respiratory signs (cough, sneezing, conjunctivitis, nasal, and/or ocular discharge).

Sera were available for 188 dogs and 63 cats for which complete signalment, history, and location were available (Fig. 1). Additional sera were collected from

diagnostic laboratories for 263 dogs and 128 cats from the affected areas, but which lacked further historical information.

Sampling of animals for this study was approved by the Ethics Committee of the Department of Veterinary Medicine, University of Bari, Italy (approval number 15/2020).

**Polymerase chain reaction.** Sample preparation and RNA extraction were carried out in the biosafety level 3 containment laboratory at the Department of Veterinary Medicine, University of Bari, Italy. Detection of SARS-CoV-2 RNA used two real-time RT-PCR assays targeting nucleoprotein and envelope protein genes[11]. Briefly, a one-step method was adopted using the Superscript III one-step RT-PCR system with Platinum Taq Polymerase (Invitrogen srl, Milan, Italy) and the following 50-μl mixture: 25 μl of master mix, 400 nM (E_Sarbeco_F, E_Sarbeco_R), 600 nM (N_Sarbeco_F), or 800 nM (N_Sarbeco_R) of primers, 200 nM of probes (E_Sarbeco_P1 or N_Sarbeco_P) (Supplementary Table 1), and 10 μl of template RNA. The thermal profile consisted of incubation at 55 °C for 10 min for reverse transcription, followed by 95 °C for 3 min and then 45 cycles of 95 °C for 15 s, 58 °C for 30 s.

**Plaque reduction neutralization test.** The SARS-CoV-2/human/Liverpool/REMRQ0001/2020 isolate was cultured in Vero E6 cells[13]. For PRNT[12], sera were heat inactivated at 56 °C for 1 h and stored at −20 °C until use. Dulbecco's minimal essential medium containing 2% fetal bovine serum and 0.05 mg/mL gentamicin was used for serial twofold dilutions of serum. SARS-CoV-2 at 800 PFU/mL was added to an equal volume of diluted serum and incubated at 37 °C for 1 h. The virus serum dilution was inoculated onto Vero E6 cells, incubated at 37 °C for 1 h, and overlaid as in standard plaque assays. Cells were incubated for 48 h at 37 °C and 5% $CO_2$ then fixed with 10% formalin and stained with 0.05% crystal violet solution. $PRNT_{80}$ was determined by the highest dilution with 80% reduction in plaques compared to the control. Samples with detectable neutralizing antibody titer were repeated as technical replicates for confirmation.

**Data analysis.** Fisher's exact test was used to analyze differences in antibody detection from households with known COVID-19 infection status, and antibody detection from male and female animals. Spearman's correlation was used to analyze the relationship between human COVID-19 case numbers and detection of antibodies in animals. All statistical analyses were performed in GraphPad Prism 6.

**Reporting summary.** Further information on research design is available in the Nature Research Reporting Summary linked to this article.

## Data availability

The authors declare that the data supporting the findings of this study are available within the article and its Supplementary Information files, or are available from the authors upon request. Source data are provided with this paper.

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

## Acknowledgements

SARS-CoV-2 RNA extracts from infected human patients that were used as positive controls in the real-time RT-PCR assays were kindly provided by Istituto Zooprofilattico della Puglia e della Basilicata, Foggia, and by Istituto Zooprofilattico dell'Abruzzo e del Molise "G. Caporale", Teramo, Italy. We are grateful to the vets that contributed to the sample collection in the COVID-19 severely affected Italian regions. This work was supported by grants of Fondazione CARIPLO-Misura a sostegno dello sviluppo di collaborazioni per l'identificazione di terapie e sistemi di diagnostica, protezione e analisi per contrastare l'emergenza Coronavirus e altre emergenze virali del futuro, project "Genetic characterization of SARS-CoV2 and serological investigation in humans and pets to define cats and dogs role in the COVID-19 pandemic (COVIDinPET)". E.I.P. was supported by the Liverpool School of Tropical Medicine Director's Catalyst Fund award. S.L.S. and A.D.R. were supported by the DogsTrust. G.L.H. was supported by the BBSRC (BB/T001240/1), the Royal Society Wolfson Fellowship (RSWF\R1\180013), NIH grants (R21AI138074 and R21AI129507), UKRI (20197), and the NIHR (NIHR2000907). G.L.H. and T.P. are affiliated to the National Institute for Health Research Health Protection Research Unit (NIHR HPRU) in Emerging and Zoonotic Infections at University of Liverpool in partnership with Public Health England (PHE), in collaboration with Liverpool School of Tropical Medicine, the University of Oxford, and the University of Liverpool. G.L.H. is based at LSTM. The views expressed are those of the author(s) and not necessarily those of the NHS, the NIHR, the Department of Health or Public Health England. The authors are grateful to all veterinary practioners that contributed to sample collection.

## Author contributions

E.I.P., A.D.R., and N.D. designed the study and wrote the first draft of the manuscript. S.L.S., A.D.R., T.P., G.T.P., and G.L.H. edited the manuscript. G.E., Al.G., and S.P. contributed to the study's design. A.D.R. helped fund the serology. C.D., S.L.S., E.R.A., T.P., E.L., M.S.L., and G.L. performed experiments. An.G., M.M., A.S., S.L., U.B., V.M., F.S.B., V.R.B., A.D.R, U.A., G.T.P., and G.L.H. analyzed data.

## Competing interests

The authors declare no competing interests.
