## [Peer Review File · Nature Communications]

Reviewers' Comments:

Reviewer #1:

Remarks to the Author:

The study aim was to answer "questions about the susceptibility of animals under natural conditions of pet ownership". To this end the authors conducted the "largest" study to investigate SARS-CoV-2 in companion animals, concluding that both cats and dogs can seroconvert under the normal conditions of pet ownership. This is not a novel conclusion, although it provides incremental support to previous findings. There have been several reports of sporadic SARS-CoV-2 infections of pet cats and dogs in which seroconversion was reported in infected animals.

The difficulties of conducting epidemiological studies of pet cats and dogs during a global pandemic are significant but the magnitude of the study, and the authors' assertions, are somewhat exaggerated.

Although clinical swabs were analysed from 340 dogs and 188 cats (all of which tested negative for SARS-CoV-2 RNA), sera were available from only 188 dogs and 63 cats for which full signalment and clinical details were known. In the Abstract, the authors state "dogs from COVID-19 positive households being significantly more likely to test positive than those from COVID-19 negative households" but as this finding was based on a comparison between 1/7 dogs from suspected COVID-19 households versus 2/133 dogs from non-COVID-19 households it is clear that additional data (testing more similar numbers of animals from COVID-19 and non-COVID-19 households) would be required to support this conclusion.

The authors assert that in "provinces where at least 10 samples were available, there was a strong positive trend between the proportion of dogs that tested positive and the recorded burden of human disease (Spearman's $r = 0.732$, $p = 0.051$) (Fig. 2)". There was a positive correlation, but the magnitude should not be overstated.

These findings provide further evidence that sporadic SARS-CoV-2 infections arise in pet dogs and cats, but the authors should modify their conclusions to recognise the limitations of the study.

Reviewer #2:

Remarks to the Author:

This manuscript describes an extensive survey of SARS-CoV-2 RNA and antibody responses in dogs and cats in Italy. This is the largest study of its kind to be conducted to date, and given the possible concerns regarding both the health of pets and their role in viral epidemiology, this means the findings of this study will be of significant interest. Overall the methodology used in this study is sound, and conclusions drawn accurately reflect the data presented. However, a number of minor comments need to be addressed as follows.

Line 85 results – was there any overlap between animals that had respiratory signs and those that were from COVID-19 positive households? It is not clear from this sentence whether there is any overlap between the groups of animals mentioned.

Line 103 – was neuter status of any of the cases known?

Line 104 – It is very interesting to see an association between the number of human cases in a region and seropositivity in animals cases, but I'm not convinced by figure 2 that this can be described as 'strong'.

Line 121 – or could this simply be reflective of the small numbers of seropositive cats identified?

Line 122 – Neuter status needs to be known before making any speculations about the apparent relevance of this finding. Neuter status would surely play an important role in any gender differences that are due to physiological or behavioural variation. Care therefore needs to be taken when comparing this data to the situation in humans. It seems most likely that apparent male bias could simply be due to sample size in this group.

Line 130 – Experimental infections of cats by Shi et al indicated that younger cats may be actually more permissive to SARS-CoV-2, when comparing cats 70-100 days old with 8 month old animals. I agree it is unknown what the susceptibility of animals >1 yr is, but it is possible that they are in fact less susceptible.

Line 148 – Low neutralising antibody titres to any virus are not always indicative of active infection, they may simply reflect exposure to significant amounts of virus (as already alluded to in the title). In the absence of identification of any infectious virus, it is difficult to claim that all animals were infected with 100% certainty, so rephrasing of this sentence would be helpful.

Line 150 – I agree it is interesting that a similar proportion of seropositive dogs were identified as cats, although the higher neutralising titres in cats could suggest this species may be more susceptible, with higher viral loads inducing more robust antibody responses.

Line 158 – spelling of holistic!

Reviewer #3:

Remarks to the Author:

This is an interesting study that investigated a relevant and understudied area.

The results are interesting but could be clearer, as could some components of the methods.

The denominator varies and this is not often clear throughout. Rather than reporting percentages, n/N should also be provided.

Line 66: Timeframe is important when referring to concurrent human disease. Specific sampling dates should be provided, at a minimum.

Sampling needs to be better described. The type of swab and transport medium should be stated. The type of nasal swab (e.g. deep, superficial) should be explained.

Why was there no standard approach to sampling for PCR? It seems like only one of nasal, oropharyngeal or rectal swabs was collected from almost all animals. This is rather unusual. How was the sampling site chosen for an individual animal?

Line 70: What was defined as “exposure to COVID-19 infected humans”. This is very important given the defined period of shedding. Exposure to an infected person a month earlier would be irrelevant in terms of the likelihood of PCR positivity, presumably.

Timing of serological testing with respect to exposure to infected people is also useful to add, since serological testing was based on neutralizing antibodies, which can wane relatively quickly.

Line 83: Rather than results at the swab level, results would be best presented at the animal level.

Line 84: “symptoms” is not an appropriate term for animals. By definition, animals cannot have

symptoms. Clinical signs would be an appropriate term.

Line 86: As above, timeframe is critical for these data. It makes little sense to include animals with exposure more than a couple weeks prior to testing.

More information about test specificity would be good, particularly given the low seroprevalence. It is stated that 'reference sera" were tested but it's not stated how many.

Line 104: What data were used for human disease?

Line 106: Specific results should be provided for the cat data, not just "association was observed".

Line 127: There is little power to support statements of younger animals getting protected, that older animals should be used in experimental models or that current models underestimate susceptibility. This should be removed.

Revision note

Ms. No. NCOMMS-20-31307-T

Evidence of exposure to SARS-CoV-2 in cats and dogs from households in Italy

Reviewer #1 (Remarks to the Author):

The study aim was to answer “questions about the susceptibility of animals under natural conditions of pet ownership”. To this end the authors conducted the “largest” study to investigate SARS-CoV-2 in companion animals, concluding that both cats and dogs can seroconvert under the normal conditions of pet ownership. This is not a novel conclusion, although it provides incremental support to previous findings. There have been several reports of sporadic SARS-CoV-2 infections of pet cats and dogs in which seroconversion was reported in infected animals.

Q: The difficulties of conducting epidemiological studies of pet cats and dogs during a global pandemic are significant but the magnitude of the study, and the authors’ assertions, are somewhat exaggerated.

R: Assertions about the magnitude of the study have been mitigated according to the reviewer’s suggestion (see throughout the text). **In addition, in the new version, we have expanded the number of samples tested by serology, so that a further 102 samples (63 dog, 39 cat) samples have been processed.**

Q: Although clinical swabs were analysed from 340 dogs and 188 cats (all of which tested negative for SARS-CoV-2 RNA), sera were available from only 188 dogs and 63 cats for which full signalment and clinical details were known. In the Abstract, the authors state “dogs from COVID-19 positive households being significantly more likely to test positive than those from COVID-19 negative households” but as this finding was based on a comparison between 1/7 dogs from suspected COVID-19 households versus 2/133 dogs from non-COVID-19 households it is clear that additional data (testing more similar numbers of animals from COVID-19 and non-COVID-19 households) would be required to support this conclusion.

R: This seems a misunderstanding of the reviewer and we apologise for not presenting data in a cleaner manner. Table 1 and the corresponding text in our manuscript (lines 96-98) clearly stated that 47 (not 7) dogs were tested from COVID-19 positive households, 6 of which (not 1) were positive for neutralizing antibodies. The reviewer seems to have confused our statements regarding dog samples obtained from COVID-19 positive households (47 total samples, 6 positive) and suspected COVID-19 positive households (7 total samples, 1 positive). **Please, note that in the modified manuscript these numbers have changed, since a further 102 (63 dog, 39 cat) samples have been processed.**

Q: The authors assert that in “provinces where at least 10 samples were available, there was a strong positive trend between the proportion of dogs that tested positive and the recorded burden of human disease (Spearman’s $r = 0.732$, $p = 0.051$) (Fig. 2)”. There was a positive correlation, but the magnitude should not be overstated.

R: The sentence has been rephrased (lines 113-115).

Q: These findings provide further evidence that sporadic SARS-CoV-2 infections arise in pet dogs and cats, but the authors should modify their conclusions to recognise the limitations of the study.

R: Conclusions have modified accordingly (throughout the Discussion).

Reviewer #2 (Remarks to the Author):

This manuscript describes an extensive survey of SARS-CoV-2 RNA and antibody responses in dogs and cats in Italy. This is the largest study of its kind to be conducted to date, and given the possible concerns regarding both the health of pets and their role in viral epidemiology, this means the findings of this study will be of significant interest. Overall the methodology used in this study is sound, and conclusions drawn accurately reflect the data presented. However, a number of minor comments need to be addressed as follows.

Q: Line 85 results – was there any overlap between animals that had respiratory signs and those that were from COVID-19 positive households? It is not clear from this sentence whether there is any overlap between the groups of animals mentioned.

R: This data have been now provided (lines 94-95).

“In addition, 64 of these dogs and 57 of the cats that tested negative were living in households previously confirmed as having had COVID-19 and 14 animals (11 cats and 3 dogs) from COVID-19 households were displaying respiratory signs at the time of sampling.”

Q: Line 103 – was neuter status of any of the cases known?

R: Information about the neuter status was available only for 1 seropositive bitch and this has been now provided (lines 112-113).

“Dogs were significantly more likely to test positive for SARS-CoV-2 neutralizing antibodies if they came from a known COVID-19 positive household (Fisher’s exact test, $p=0.004$) or were male (Fisher’s exact test, $p=0.045$), while there was insufficient data to assess any correlation with the neuter status, since this was reported only for 1 seropositive bitch.”

Q: Line 104 – It is very interesting to see an association between the number of human cases in a region and seropositivity in animals cases, but I’m not convinced by figure 2 that this can be described as ‘strong’.

R: The sentence has been rephrased according to the reviewer’s suggestion (line 114).

Q: Line 121 – or could this simply be reflective of the small numbers of seropositive cats identified?

R: A sentence about this limitation has been added to the text (lines 132-133).

“This contrasts experimental studies where dogs were less susceptible to infection⁹. However, this finding could be also reflective of the small numbers of seropositive cats identified.”

Q: Line 122 – Neuter status needs to be known before making any speculations about the apparent relevance of this finding. Neuter status would surely would play an important role in any gender differences that are due to physiological or behavioural variation. Care therefore needs to be taken when comparing this data to the situation in humans. It seems most likely that apparent male bias could simply be due to sample size in this group.

R: The sentence has been rephrased according to the reviewer’s suggestion (lines 134-136).

“In addition, a higher proportion of male dogs were seropositive compared to female dogs, but also in this case the results could have been biased by the small size of positive animals. In fact, neuter status was unknown for the majority of seropositive animals, which prevented a full comparison with the situation in humans.”

Q: Line 130 – Experimental infections of cats by Shi et al indicated that younger cats may be actually more permissive to SARS-CoV-2, when comparing cats 70-100 days old with 8 month old animals. I agree it is unknown what the susceptibility of animals >1 yr is, but it is possible that they are in fact less susceptible.

R: The sentence has been rephrased according to the reviewer’s suggestion (lines 142-144).

“However, although the experimental susceptibility of adult pets is still unknown, in cats aged less than 1 year a higher susceptibility to SARS-CoV-2 infection was observed at 70-100 days than at 8 months of age⁹.”

Q: Line 148 – Low neutralising antibody titres to any virus are not always indicative of active infection, they may simply reflect exposure to significant amounts of virus (as already alluded to in the title). In the absence of identification of any infectious virus, it is difficult to claim that all animals were infected with 100% certainty, so rephrasing of this sentence would be helpful.

R: The sentence has been rephrased according to the reviewer's suggestion (lines 162-163).

"In this extensive epidemiological survey of SARS-CoV-2, we found that companion animals living in areas of high human infection have been exposed to SARS-CoV-2, thus confirming previous reports of natural infections of dogs and cats with the novel coronavirus."

Q: Line 150 – I agree it is interesting that a similar proportion of seropositive dogs were identified as cats, although the higher neutralising titres in cats could suggest this species may be more susceptible, with higher viral loads inducing more robust antibody responses.

R: The sentence has been rephrased according to the reviewer's suggestion (lines 164-166).

"Our results suggest that dogs warrant further investigation regarding SARS-CoV-2 susceptibility, although higher antibody titers were detected in cats, which is in agreement with previous studies suggesting that these animals are most susceptible to SARS-CoV-2 infection⁹."

Q: Line 158 – spelling of holistic!

R: Corrected (line 174).

Reviewer #3 (Remarks to the Author):

This is an interesting study that investigated a relevant and understudied area.

The results are interesting but could be clearer, as could some components of the methods.

R: Methods and results have been made clearer (see below).

R: The denominator varies and this is not often clear throughout. Rather than reporting percentages, n/N should also be provided.

Q: These data have been now provided (lines 96-97).

Q: Line 66: Timeframe is important when referring to concurrent human disease. Specific sampling dates should be provided, at a minimum.

R: This information has been provided (lines 63).

"All animals were sampled by their private veterinary surgeon during routine healthcare visits between March 15 and May 11, 2020 (Supplementary material Data S1)."

Q: Sampling needs to be better described. The type of swab and transport medium should be stated. The type of nasal swab (e.g. deep, superficial) should be explained.

R: More details about sampling methods have been provided (lines 70-75).

"Oropharyngeal (306 dogs, 175 cats), nasal (185 dogs, 77 cats), and/or rectal (66 dogs, 30 cats) swabs were collected from a total of 494 pets using synthetic fiber swabs. Oropharyngeal swabs were collected by inserting the swab into the posterior pharynx and tonsillar areas. For collection of nasal secretions, the same swab was inserted into each nostril and rotated in order to be saturated by the nasal fluid. Rectal swabs were collected by inserting the swab 1-2 cm past the anal verge and rotating the swab gently 360°. After swabbing each swab was immersed in 1.5 ml of viral transport medium."

Q: Why was there no standard approach to sampling for PCR? It seems like only one of nasal, oropharyngeal or rectal swabs was collected from almost all animals. This is rather unusual. How was the sampling site chosen for an individual animal?

R: It has been clarified that more than one sample type were collected from each animal (lines 69-70, 91). In fact, the number of animals tested by real-time PCR were 494 (line 70) and the total number of swabs tested were 841 (information removed from the text according to the reviewer's suggestion).

Q: Line 70: What was defined as "exposure to COVID-19 infected humans". This is very important given the defined period of shedding. Exposure to an infected person a month earlier would be irrelevant in terms of the likelihood of PCR positivity, presumably.

R: This information has been provided (line 77).

"For 340 dogs and 188 cats, full signalment and clinical history were available, including breed, sex, age, exposure to COVID-19 infected humans in the previous 2 weeks (COVID-19 positive household, suspected COVID-19 positive household but not confirmed by specific assay, and COVID-19 negative household), presence of respiratory signs (cough, sneezing, conjunctivitis, nasal and/or ocular discharge)."

Q: Timing of serological testing with respect to exposure to infected people is also useful to add, since serological testing was based on neutralizing antibodies, which can wane relatively quickly.

R: This information has been provided (line 77).

Q: Line 83: Rather than results at the swab level, results would be best presented at the animal level.

R: Modified as suggested (line 91).

Q: Line 84: "symptoms" is not an appropriate term for animals. By definition, animals cannot have symptoms. Clinical signs would be an appropriate term.

R: Corrected as suggested throughout the text.

Q: Line 86: As above, timeframe is critical for these data. It makes little sense to include animals with exposure more than a couple weeks prior to testing.

R: This information has been provided (lines 93-94).

"In addition, 64 of these dogs and 57 of the cats that tested negative were living in households previously confirmed as having had COVID-19 and 14 animals (11 cats and 3 dogs) from COVID-19 households were displaying respiratory signs at the time of sampling."

Q: More information about test specificity would be good, particularly given the low seroprevalence. It is stated that 'reference sera' were tested but it's not stated how many.

R: This information has been provided (lines 107-109).

Q: Line 104: What data were used for human disease?

R: data used for human disease were provided with Supplementary Material Data S1. This has been now added to the text (line 115).

Q: Line 106: Specific results should be provided for the cat data, not just "association was observed".

R: Modified as suggested (lines 117-118).

Q: Line 127: There is little power to support statements of younger animals getting protected, that older animals should be used in experimental models or that current models underestimate susceptibility. This should be removed.

R: The sentence has been removed.

Reviewers' Comments:

Reviewer #1:

Remarks to the Author:

The detailed point-by-point rebuttal was appreciated and the requested revisions of the manuscript have been made.

Reviewer #2:

Remarks to the Author:

The authors have addressed all comments I raised to my satisfaction.

The only point that could be extended upon regards the age of experimentally infected animals - Bosco-Lauth et al have very recently published the results of experimental infections in 7 adult cats. I appreciate this resubmission was likely made before the results of this study were available, but if possible it would be good to include a comment about the lack of clinical disease in adult cats as this was a point of discussion in the manuscript.

Reviewer #3:

Remarks to the Author:

I appreciate the efforts that were put into this revision. I have a couple minor questions or suggestions, focusing on the data.

Previous response: It has been clarified that more than one sample type were collected from each animal (lines 69-70, 91). In fact, the number of animals tested by real-time PCR were 494 (line 70) and the total number of swabs tested were 841 (information removed from the text according to the reviewer's suggestion).

I'm still a bit unclear. The cited text mentions and/or in terms of swab collection, so it is not clear to the reader if multiple samples were collected from all. It's still unclear why there were differences in sample types (why were some sites sampled from some animals but not others). Maybe a table would help.

Line 90: Did they truly all have respiratory 'distress', a fairly rare and serious presentation of respiratory tract disease? Or, should this be 'respiratory disease' not 'respiratory distress'.

Why are results from the suspected households not presented in the table?

It would be useful to report the total # of households for dogs and cats too, unless there was only one sampled per household (and that could be stated). If there were multiple samples per household, the effect of household should be controlled for in the statistical analysis since all samples would not be independent.

Line 76: Was a specific yes/no answer available for the exposure to COVID+ people question? (Was this queried specifically or was it dependent on someone noting previous exposure? I assume all owners were queried but the method mention data that 'were available' suggesting secondary use of information vs specific questioning. This is relevant to have confidence in the classifications. If it is based on secondary information, the data are fine, but this should be discussed as a limitation.

Revision note

Manuscript NCOMMS-20-31307A-Z

Evidence of exposure to SARS-CoV-2 in cats and dogs from households in Italy

Reviewer #1 (Remarks to the Author):

Q: The detailed point-by-point rebuttal was appreciated and the requested revisions of the manuscript have been made.

R: We thank you for your previous suggestions.

Reviewer #2 (Remarks to the Author):

Q: The authors have addressed all comments I raised to my satisfaction.

R: We thank you for your previous suggestions.

Q: The only point that could be extended upon regards the age of experimentally infected animals - Bosco-Lauth et al have very recently published the results of experimental infections in 7 adult cats. I appreciate this resubmission was likely made before the results of this study were available, but if possible it would be good to include a comment about the lack of clinical disease in adult cats as this was a point of discussion in the manuscript.

R: Thank you for the suggestion. This reference has added important information to our discussion points. We have updated the discussion to include this study in the manuscript (lines 148-149 and ref. no. 18).

Reviewer #3 (Remarks to the Author):

Q: I appreciate the efforts that were put into this revision. I have a couple minor questions or suggestions, focusing on the data.

R: We thank you for your previous suggestions.

Q: Previous response: It has been clarified that more than one sample type were collected from each animal (lines 69-70, 91). In fact, the number of animals tested by real-time PCR were 494 (line 70) and the total number of swabs tested were 841 (information removed from the text according to the reviewer's suggestion).

I'm still a bit unclear. The cited text mentions and/or in terms of swab collection, so it is not clear to the reader if multiple samples were collected from all. It's still unclear why there were differences in sample types (why were some sites sampled from some animals but not others). Maybe a table would help.

R: Thank you for the suggestion. We have added a table (new Table 1) in order to summarize the different sample types collected from dogs and cats. We planned to collect at least the oropharyngeal swab from each animal and we were successful in the majority of samples pets. When possible, we also asked vets to collect the nasal and rectal swabs.

Q: Line 90: Did they truly all have respiratory ‘distress’, a fairly rare and serious presentation of respiratory tract disease? Or, should this be ‘respiratory disease’ not ‘respiratory distress’.

R: Corrected as suggested (line 95).

Q: Why are results from the suspected households not presented in the table?

R: The suspected household samples have been added to Table 2. They were originally excluded from the table because they did not fit in either COVID+ or COVID- category for the statistical analysis.

Q: It would be useful to report the total # of households for dogs and cats too, unless there was only one sampled per household (and that could be stated). If there were multiple samples per household, the effect of household should be controlled for in the statistical analysis since all samples would not be independent.

R: There was only one animal sampled per household and this has been now stated (lines 81-82).

Q: Line 76: Was a specific yes/no answer available for the exposure to COVID+ people question? (Was this queried specifically or was it dependent on someone noting previous exposure? I assume all owners were queried but the method mention data that ‘were available’ suggesting secondary use of information vs specific questioning. This is relevant to have confidence in the classifications. If it is based on secondary information, the data are fine, but this should be discussed as a limitation.

R: For pets that underwent both serological and molecular analyses (340 dogs and 188 cats) owners were specifically asked about COVID positive households, and there was a specific question in the signalment/anamnesis form. We have now changed “were available” to “were collected” for more clarity (line 77).